# MEMORY OPTIMIZATION FOR DEEP NETWORKS

**Aashaka Shah[1], Chao-Yuan Wu[1], Jayashree Mohan[1], Vijay Chidambaram[1,2], Philipp Krähenbühl[1]**
[1]University of Texas at Austin
[2]VMware Research
{aashaka,cywu,jaya,vijay,philkr}@cs.utexas.edu

## ABSTRACT

Deep learning is slowly, but steadily, hitting a memory bottleneck. While the tensor computation in top-of-the-line GPUs increased by $32\times$ over the last five years, the total available memory only grew by $2.5\times$. This prevents researchers from exploring larger architectures, as training large networks requires more memory for storing intermediate outputs. In this paper, we present MONET, an automatic framework that minimizes both the memory footprint and computational overhead of deep networks. MONET jointly optimizes the checkpointing schedule and the implementation of various operators. MONET is able to outperform all prior hand-tuned operations as well as automated checkpointing. MONET reduces the overall memory requirement by $3\times$ for various PyTorch models, with a 9-16% overhead in computation. For the same computation cost, MONET requires 1.2-1.8$\times$ less memory than current state-of-the-art automated checkpointing frameworks. Our code is available at https://github.com/utsaslab/MONeT.

## 1 INTRODUCTION

Deep networks are widely used in domains ranging from image classification (Krizhevsky et al., 2012; Simonyan & Zisserman, 2015; He et al., 2016) to video recognition (Wu et al., 2019; Feichtenhofer et al., 2019) or natural language processing (Devlin et al., 2019; Yang et al., 2019). However, training deep networks is resource-intensive. In particular, the amount of GPU memory bottlenecks training many deep networks (Dong et al., 2016; Kim et al., 2016; Chen et al., 2018; Child et al., 2019). This bottleneck requires either modifying the network architecture or scaling training to multiple nodes, incurring significant overheads.

We presents MONET, an automatic framework to minimize memory footprint for deep networks. MONET *jointly* optimizes global compute-graph-level techniques (such as checkpointing) and local techniques (such as memory-efficient implementations of individual operator). At the heart of MONET is a theoretical analysis that enables joint optimization and provides tight bounds on memory consumption. We analyze the memory consumption and computational cost of a general forward and backward pass under changing local operator implementations and a global checkpointing schedule. Specifically, we are able to tightly bound the peak memory consumption for network forward, backward, and recomputation stages. MONET uses these constraints to optimize for the most efficient forward and backward implementation both locally and globally under a fixed memory budget. We linearize all memory bounds, and express both implementation selection and checkpointing as a 0-1 integer program, which we solve using standard solvers.

We conduct extensive experiments, demonstrating that MONET significantly outperforms existing automatic frameworks that use local or global techniques. On multiple architectures (ResNet (He et al., 2016), VGG (Simonyan & Zisserman, 2015), UNet (Ronneberger et al., 2015), GoogleNet (Szegedy et al., 2015), MobileNet-V2 (Sandler et al., 2018)), memory budgets (5-10 GB), and network configurations (multiple resolutions), MONET consistently achieves lower memory footprints at equivalent or lower computational overhead. MONET reduces the overall memory requirement by $3\times$ for various models, with a 9-16% overhead in computation. For the same computation cost, MONET requires 1.2-1.8$\times$ less memory than the current state-of-the-art automated checkpointing framework. The results achieved by MONET demonstrate the power of jointly optimizing global checkpointing schedules and local operator implementations.

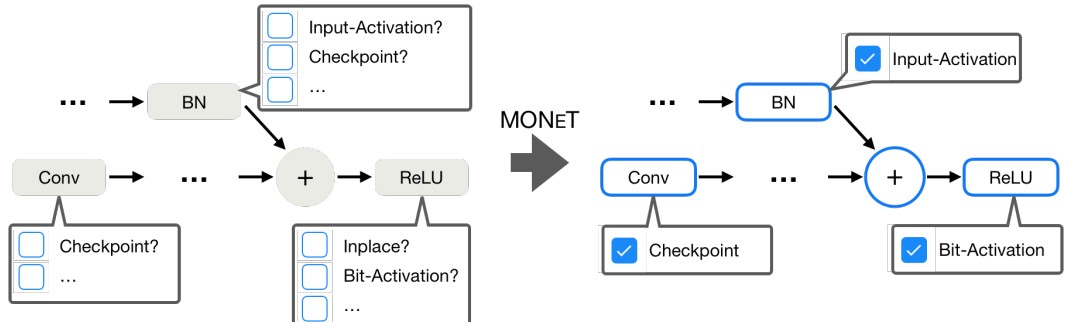

Figure 1: **Memory Optimized Network Training (MONeT)**, an automatic framework that minimizes the memory footprint of deep networks by jointly optimizing global and local techniques.

## 2    RELATED WORK

There are two broad families of approaches to reduce the memory footprint of a deep network during training: operator-level implementation changes, and global, graph-level optimizations. The novel aspect of MONeT is that it is able to combine both approaches and find the optimal mix of local and global techniques for a given network.

**Operator-Specific Optimizations.**  Researchers have found creative ways to implement individual operators or groups of operators in a more memory-efficient manner.  Standard deep learning frameworks (Jia et al., 2014; Collobert et al., 2011; Paszke et al., 2019; Abadi et al., 2016) provide different implementations of certain operators that trade computation for intermediate memory use. These implementation are chosen according to local search heuristics, and are not globally optimal. Gist (Jain et al., 2018) proposes several hand-crafted optimizations such as storing only ReLU signs. RevNets (Gomez et al., 2017) redesigns a ResNet (He et al., 2016) architecture making each network block reversible, thereby eliminating the need to store intermediate activations for backpropagation. Memory-efficient DenseNets (Pleiss et al., 2017) reduce memory utilized for feature maps by recomputing all intermediate feature maps during the backward pass with a small compute overhead. In-place activated batchnorm (Bulò et al., 2018) or ReLU layers use output activations to compute their gradients, thus reusing a single memory buffer for the gradient computation in consecutive layers. Mixed precision training (Micikevicius et al., 2018) uses half precision (FP16) instead of single precision (FP32) for all tensors and arithmetic during training, reducing the memory by nearly half. While training at precision lower than FP16 results in loss of training quality (Banner et al., 2018), prior work like backpropagation with approximate activations (Chakrabarti & Moseley, 2019) carefully quantize certain intermediate outputs (activations) to 4 bits, resulting in significant memory savings.  Although these hand-crafted techniques independently result in memory savings, there is no one-size-fits-all recipe, and different implementations perform best on different architectures. In contrast, MONeT automatically finds the best implementation for each forward and backward operator given a memory budget.

**Checkpointing.** Chen et al. (2016) proposed dividing a network into different segments, dropping all intermediate outputs within each segment, and recomputing them later. Chen *et al.* use $\sqrt{n}$ equal segments, trading memory savings for the cost of an extra forward pass. Checkmate (Jain et al., 2019) solves the problem in a more general setting, using an mixed-integer linear program solver to decide which layers to recompute for a given network.  Like Checkmate, our work optimizes a checkpointing schedule, but on a different computation graph.  Our computation graph allows for the optimization of an entire execution plan jointly finding a checkpointing schedule and the best implementation of each forward and backward operator.  In Checkmate, changes in operator implementation induce a different computation graph, and could thus not directly be optimized. Appendix F highlights some of the difficulties of adding operator optimizations into Checkmate.

In summary, while much work has been done on local optimizations (operator implementations) and global compute-graph-level techniques (automated checkpointing), MONeT is the first system to jointly optimize a given architecture using both local and global techniques.

---

**Algorithm 1:** Forward Pass

**Input** : Inputs, $\theta$, a schedule $(s, r)$.
**Output:** Output tensor

1   $S^N = \{\}$    /* Saved tensors for backward */
2   $L = \{\text{inputs}, \theta\}$ /* Local tensors for forward */

3   **for** $i = 1 \ldots N$ **do**
4     $x_i = \text{forward}_i(L)$

5     Add $x_i$ to $L$
6     Remove all tensors from $L$ that are not used later

7     **if** $s_i^N$ **then**
8      Add $x_i$ to $S^N$

9   **return** $L$

---

**Algorithm 2:** Backward Pass

**Input** : Loss gradients, inputs, $\theta$, $S^N$, $(s, r)$.
**Output:** Output tensor

1   $\hat{L} = \{\text{loss gradients}\}$ /* Local backward tensors */
2   **for** $k = N \ldots 1$ **do**
3     $L = S^k$     /* Local forward tensors */
4     $S^{k-1} = \{\}$      /* Saved tensors */
5     **for** $i = 1 \ldots N$ **do**
6      **if** $r_i^k$ **then**
7       $x_i = \text{forward}_i(L)$
8       Add $x_i$ to $L$
9      Remove all tensors from $L$ not used later
10      **if** $s_i^{k-1}$ **then**
11       Add $x_i$ to $S^{k-1}$    /* use $x_i \in L$ */
12     $y_k = \text{backward}_k(\hat{L}, L)$
13     Add $y_k$ to $\hat{L}$
14     Remove tensors from $\hat{L}$ that are not used later

---

Figure 2: **Schematic overview of the forward and backward passes.** The algorithms include aggressive memory savings by greedily freeing unused tensors, and allow for a general checkpointing schedule $(s, r)$ to be executed.

## 3   PRELIMINARIES

Let the forward pass of a CNN with parameters $\Theta$ be expressed as a directed-acyclic graph (DAG), where each node $i \in \{1, \ldots, N\}$ corresponds to an operator $\text{forward}_i$, and edges $(i, j) \in \mathbf{E}$ specify the data-flow dependencies, *i.e.,* the output of operator $i$ is used as input in operator $j$. Without loss of generality, computational dependency $(i, j) \in \mathbf{E}$ implies $i < j$. Let $\mathcal{N}_j = \{i : (i, j) \in \mathbf{E}\}$ be the set of all incoming edges of an operation $j$.

We will first discuss the forward pass through a network and the basic form of a backward pass using checkpointing. The backward pass reverses all computational dependency expressed in our DAG, and induces certain dependencies on forward activations. We call these checkpoint dependencies $\mathcal{D}_k$. They are either saved or recomputed depending on a schedule $(s, r)$. Checkpointing creates a trade-off between computation and memory consumption. To highlight this trade-off, we formally compute the amount of memory consumed in both forward and backward passes, which allows us to optimize for the ideal execution plan in Sec. 4. We provide a reference to the notations introduced in this section and the next along with their explanations in Appendix A.

**The Forward Pass.** Alg. 1 shows a general overview of the forward pass in a deep network, as implemented in standard deep learning frameworks (Jia et al., 2014; Collobert et al., 2011; Paszke et al., 2019; Abadi et al., 2016). The algorithm proceeds in increasing order of index $i$. Each operator $\text{forward}_i(\cdot)$ depends on a set of tensors $L$ stored in local memory. These tensors include model parameters $\Theta$, computational dependencies $\mathcal{N}_i$, and tensors stored for later forward operators, i.e. skip or residual activations (He et al., 2016). At each iteration, we add any output tensors of $\text{forward}_i$ to the local memory $L$. Early deep learning frameworks (Jia et al., 2014; Collobert et al., 2011) strictly grew the set of local tensors $L$ leading to an unnecessarily high memory consumption. Modern graph-based frameworks (Paszke et al., 2019; Abadi et al., 2016) reduce the memory footprint by aggressively pruning local memory $L$ and freeing any tensor that is no longer used in later computations. Some output activations $x_i$ are used in the backward pass, and have to be saved for later. We use a checkpointing schedule $s^N$ to determine which. Formally, $s_i^N \in \{0, 1\}$ indicates whether the activation of node $i$ is stored during the forward pass. An activation which is not stored will be recomputed if it is needed during the backward pass.

**Analyzing peak memory consumption of the forward pass.** Only the $\text{forward}_i$ operator (Alg. 1 L. 4) allocates memory. All other operators perform mere bookkeeping on existing tensor. It is thus sufficient to study the peak memory consumption $m_i^N$ in $\text{forward}_i$ for each node $i$. Let $L_i, S_i^N$ be the set of local tensors $L$ and saved tensors $S$ while calling $\text{forward}_i$ respectively. $L_i$ includes all parameters and computational dependencies for this and later forward passes $L_i = \Theta \cup \{x_j : j \in \mathcal{N}_t \text{ for any } t \geq i \text{ and } j < i\}$. $L_i$ is constant and computed ahead of time. The schedule $s^N$ determines the set of saved tensors $S_i^N = \{x_j : s_j^N = 1 \text{ for } j < i\}$. In addition, each forward

operator uses a certain amount of workspace memory $c_i$ to store intermediate results. The total memory consumption of a forward operator is thus

$$m_i = c_i + |x_i| + |S_i^N \cup L_i| = c_i + |x_i| + \sum_{x_j \in L_i} |x_j| + \sum_{j < i : x_j \notin L_i} |x_j| s_j^N, \qquad (1)$$

where $| \cdot |$ refers to the memory consumed by a tensor or set of tensors. Most of the memory consumption is constant and does not depend on the schedule.

**The Backward Pass.** The backward pass proceeds in a reverse order, as summarized in Alg. 2. $\mathrm{backward}_k(\cdot)$ of each node $k$ depends on a set of gradient tensors $\hat{L}$ and forward tensors $\{x_i : i \in \mathcal{D}_k\}$. Any gradients required by the current and later backward passes are stored in local memory $\hat{L}$. Dependencies $\mathcal{D}_k$ may either be stored in $S^k$ or need to be recomputed from checkpoints in $S^k$. Recomputation involves forward computation of one or more nodes, which increases computational overhead, and allows for a new set of tensors $S^{k-1}$ to be saved. After recomputation, all dependencies $\mathcal{D}_k$ are kept in memory. The backward operation produces a gradient for each input tensor of the original forward operation, which is added to $\hat{L}$ if required for a later backward computation. We aggressively remove tensors in $\hat{L}$ that are not required.

**Analyzing the peak memory consumption of the backward pass.** Peak memory consumption $\hat{m}_k$ again only depends on the $\mathrm{forward}_i$ (Alg. 2 L. 7) and $\mathrm{backward}_k$ (Alg. 2 L. 12) operations. For the $\mathrm{backward}_k$ operation, let $\hat{c}_k$ be the workspace memory, $\hat{L}_k$ be the set of gradient tensors stored, $D_k = \{x_i : i \in \mathcal{D}_k\}$ be the forward tensors used, and $S^{k-1}$ be the set of newly saved tensors. Here $\hat{L}_k$ and $D_k$ can be pre-computed. The total memory consumption for the $\mathrm{backward}_k$ call is

$$\hat{m}_k = \hat{c}_k + |y_k| + |S^{k-1} \cup \hat{L}_k \cup D_k| = \hat{c}_k + |y_k| + \sum_{y_l \in \hat{L}_k} |y_l| + \sum_{x_i \in D_k} |x_i| + \sum_{x_i \notin D_k} s_i^{k-1} |x_i|. \quad (2)$$

Here again, only the last term depends on the checkpointing schedule, while the rest is a constant.

**Analyzing the peak memory consumption of the recomputation.** Finally, the peak memory $\tilde{m}_i^k$ for the $\mathrm{forward}_i$ call (Alg. 2 L. 7) depends on the set of local tensors $L$, checkpoint dependencies $D$, saved tensors $S$, and gradient tensors $\hat{L}$, named $L_i^k$, $D_k$, $S_i^{k-1}$, $\hat{L}_k$ respectively. Following the forward pass:

$$\tilde{m}_i^k = c_i + |x_i| + |\hat{L}_k| + |S_i^{k-1} \cup L_i^k \cup D_k|$$
$$= c_i + |x_i| + |\hat{L}_k| + \sum_{j < i : x_j \notin L_i^k \cup D_k} s_j^{k-1} |x_j| + \sum_{j < i : x_j \in L_i^k \cup D_k} |x_j| + \sum_{j > i} s_j^k |x_j|. \quad (3)$$

Unlike the forward pass, $L_i^k$ is no longer constant, but depends on past saved tensors and future recomputations in $(s, r)$: $L_i^k = \Theta \cup \{x_j : j \in \mathcal{N}_t$ for any $t \geq i$ with $r_t^k = 1$ and $j < i\}$.

In the next section, we show how to take this formalization of the forward and backward pass, and find an optimal execution plan including checkpointing schedule $(s, r)$, $\mathrm{forward}_i$ implementations, and $\mathrm{backward}_k$ implementations, under a fixed memory budget.

## 4 METHOD

Our goal is to find a global checkpointing schedule $(s, r)$ and local $\mathrm{forward}_i$/$\mathrm{backward}_k$ implementations that jointly minimize the computation cost $\tau$ within a memory budget $M$. We show how to express this optimization in a 0-1 integer program and efficiently solve it. To this end, we linearize any peak memory consumption constraints, ensure that the checkpointing schedule is valid, and solve to minimize a computation cost objective. We keep track of the three contributors to memory and computational cost - forward pass, backward pass, and recomputation of forward operators.

**Memory Constraints.** Consider the case of basic checkpointing using only a single implementation for $\mathrm{forward}_i$ and $\mathrm{backward}_k$. The memory consumption of the forward 1 and backward 2 pass are linear in $s$, and thus efficiently expressed in an integer program. However, recomputation depends both on $s^{k-1}$ and $r^k$ in a non-linear manner through the local memory $L_i^k$. This joint dependence on optimization variables gives rise to quadratic constraints, which cannot directly be incorporated into

an integer program. For simplicity in this derivation, we bound the set of local tensors from above, assuming every future tensor is recomputed. We give more information about this in Appendix B. The upper bound $\bar{L}_i^k$ is constant, yielding a linear upper bound $\bar{m}_i^k$ of the recomputation memory $\tilde{m}_i^k$ analogous to Eq. 3. The set of memory constraints is thus

$$m_i \leq M \quad \forall_i \qquad \text{and} \qquad \hat{m}_k \leq M \quad \forall_k \qquad \text{and} \qquad \bar{m}_i^k \leq M \quad \forall_{k,i} \tag{4}$$

To enable operator optimization, we use a bit-vector $\delta$ to indicate the selection of an operator implementation. We add $\delta$ to the constraints which allows us to jointly optimize checkpointing $(s, r)$ and operator implementations $\delta$.

**Forward Operator Optimization.** Let each forward operator $\text{forward}_i$ have multiple different implementations $\mathcal{I}_i = \{a, b, c, \dots\}$. For examples, convolution may be implemented using matrix multiplication, the Winograd algorithm (Winograd, 1980), a Fourier transform, etc. (Chetlur et al., 2014). All implementations follow the same DAG structure, and thus use the same dependencies $\mathcal{N}_i$. However, each implementation trades workspace memory $\{c_i^a, c_i^b, \dots\}$ for computational efficiency $\{\tau_i^a, \tau_i^b, \dots\}$ in a different manner. Our experiments show that this trade-off is often complex.

Our goal is to represent the peak memory when using multiple $\text{forward}_i$ implementations in the forward pass and recomputation. Let $\delta_{i,a} \in \{0, 1\}$ indicate that implementation $a \in \mathcal{I}_i$ is used for $\text{forward}_i$ in the forward pass. Each forward operator should use exactly one implementation $\sum_l \delta_{i,l} = 1$. The choice of implementation determines the operator's computational cost $\sum_l \tau_i^l \delta_{i,l}$ and workspace memory $c_i = \sum_l c_i^l \delta_{i,l}$. Analogously, each recomputation of $\text{forward}_i$ during $\text{backward}_k$ chooses between implementations $\delta_{i,a}^k \in \{0, 1\}$ when needed $\sum_l \delta_{i,l}^k = r_i^k$, with equivalent cost estimates $\sum_l \tau_i^l \delta_{i,l}^k$ and workspace memory use $c_i^k = \sum_l c_i^l \delta_{i,l}^k$. In this formulation, all additional memory requirements remain linear and are directly integrated into the linear memory constraints or their linear relaxations (equation 4).

**Backward Operator Optimization.** Let each backward operator $\text{backward}_k$ have a set of different implementations $\hat{\mathcal{I}}_k = \{a, b, c, \dots\}$. Each implementation again trades workspace memory $\{\hat{c}_k^a, \hat{c}_k^b, \dots\}$ for computational cost $\{\hat{\tau}_k^a, \hat{\tau}_k^b, \dots\}$. While gradient tensors follow the fixed DAG structure, different implementations may depend on different forward activations $\{\mathcal{D}_k^a, \mathcal{D}_k^b, \dots\}$. For example, in-place activated operators (Bulò et al., 2018) depend on their output activation, while regular operators use the input activation. This change in the dependency structure makes optimizing for backward-operator implementations challenging.

We again aim to represent memory in terms of implementations for each $\text{backward}_k$ operator. Let $\hat{\delta}_{k,a} \in \{0, 1\}$ indicate that implementation $a \in \hat{\mathcal{I}}_k$ is used at node $k$ in the backward pass. Each backward operator should use exactly one implementation $\sum_l \hat{\delta}_{k,l} = 1$, with a computational cost $\sum_l \hat{\tau}_k^l \hat{\delta}_{k,l}$ and workspace memory $\hat{c}_k = \sum_l \hat{c}_k^l \hat{\delta}_{k,l}$. The workspace memory adds a linear constraint to the memory consumption $\hat{m}_k$ equation 2.

The biggest changes to the optimization problem, comes from the *changing dependency structure*. $\mathcal{D}_k$ is no longer constant. Instead, the implementation of a backward operator changes the set of computational dependencies $D_k$ obtained from $\mathcal{D}_k^l$. To deal with this changing dependency structure, we use the indicator vector $\hat{\delta}_k$ to select memory contribution of dependencies from the chosen implementation. This changes the backward memory consumption to

$$\hat{m}_k = \underbrace{\sum_l \hat{c}_k^l \hat{\delta}_{k,l}}_{\hat{c}_k} + |y_k| + |\hat{L}_k| + \sum_l \hat{\delta}_{k,l} \cdot |D_k^l \cup S^{k-1}|, \tag{5}$$

and the corresponding peak recomputation memory $\bar{m}_i^k$ to

$$\bar{m}_i^k = c_i + |x_i| + |\hat{L}_k| + \sum_l \hat{\delta}_{k,l} \cdot |S_i^{k-1} \cup \bar{L}_i^k \cup D_k^l|. \tag{6}$$

Note, the last term of equation 5 and equation 6 are quadratic in the original optimization variables $s_i^{k-1}$, which determines $S^{k-1}$, and $\hat{\delta}_{k,l}$. However, for binary variables, it can be linearized using an auxiliary variable (see Appendix C.4). We show the full equation expansion in Appendix C.1.

**Checkpointing Constraints.** The computational dependencies of forward and backward operators impose strict constraints on the checkpointing schedule. Any schedule violating these constraints cannot be executed. Recomputation $r_i^k$ requires saved $s_j^{k-1}$ or recomputed $r_j^k$ dependencies $j \in \mathcal{N}_i$, and only previously stored or recomputed tensors can be saved:

$$r_i^k \leq s_j^{k-1} + r_j^k \quad \forall_{i,k,j \in \mathcal{N}_i} \qquad \text{and} \qquad s_i^{k-2} \leq s_i^{k-1} + r_i^k \quad \forall_{i,k}. \qquad (7)$$

Furthermore, all forward tensors $\mathcal{D}_k^l$ required by backward$_k$ need to be stored or computed

$$s_i^{k-1} + r_i^k \geq \hat{\delta}_{k,l} \quad \forall_{k,l,i \in \mathcal{D}_k^l}. \qquad (8)$$

**Objective.** Our goal is to minimize the amount of computation required for the forward and backward pass. This is represented as the sum of computational costs of all operators:

$$\underbrace{\sum_i \sum_l \tau_i^l \delta_{i,l}}_{\text{forward pass}} + \underbrace{\sum_k \sum_l \hat{\delta}_{k,l} \hat{\tau}_k^l}_{\text{backward pass}} + \underbrace{\sum_k \sum_l \tau_i^l \delta_{i,l}^k}_{\text{recomputation}}. \qquad (9)$$

Objective equation 9 with constraints equation 4, equation 7, equation 8, and definitions equation 1, equation 5, equation 6 form our final optimization objective. It jointly solves for the optimal implementation of each forward and backward operator, as well as an efficient checkpointing schedule.

## 5 EXPERIMENTS

**Implementation Details.** We develop MONET in PyTorch v1.5.1 and solve the joint optimization problem using the Gurobi (2014) solver. Appendix D provides more implementation details and a full list of optimized operators.

The UNet experiments use $608 \times 416$ inputs following prior work (Jain et al., 2019). All other experiments use $224 \times 224$ inputs following conventions (Krizhevsky et al., 2012; Simonyan & Zisserman, 2015; He et al., 2016). Batch size for the experiments is fixed to be the maximum at which the model can be trained using baseline PyTorch on a 16 GB GPU. Since Checkmate's (Jain et al., 2019) execution engine is built for TensorFlow, and an official Gist (Jain et al., 2018) implementation is not available, we reimplement them in PyTorch for our comparisons. Our Checkmate implementation is competitive, it uses the original Checkmate solver and has the same network structure as MONET. Checkmate does not optimize for operator implementations like convolutions, so we show its runtime using the default convolution algorithm (Checkmate-D). For a stronger comparison, we also show the runtime of a Checkmate schedule that is post-optimized to greedily run the fastest convolution algorithm (Checkmate-O). Wherever not explicitly specified, we compare with Checkmate-O. All checkpointing schedules are run using the same software implementations and costs are profiled on the same hardware (NVIDIA P100 GPUs). In order to compare against operator-specific optimizations, we reimplement all Gist techniques in PyTorch and run them on our execution engine. See Appendix E for more details about our baseline implementations.

**Detailed Comparison to Baselines.** (a) *Checkpointing*: Table 1 compares the memory savings obtained by MONET and Checkmate for five different models when computational overhead over PyTorch is fixed to be 10%. MONET schedules use **2-3**× less memory than PyTorch. For the same computational overhead, MONET uses 1.2-1.8× less memory than Checkmate.

Fig. 3 shows more detailed runtime-memory trade-offs of MONET to PyTorch and Checkmate for different models. We plot the average iteration time of training as % overhead over PyTorch for

|  | ResNet-50 | GoogleNet | UNet | VGG-16 | MobileNet-V2 |
|---|---|---|---|---|---|
| PyTorch | 15.1 | 14.9 | 14.3 | 14.1 | 14.5 |
| Checkmate (Jain et al., 2019) | 8.2 | 10.5 | 9.1 | 9.9 | 5.8 |
| **MONeT** | **5.7** | **6.9** | **5.2** | **5.5** | **4.8** |

Table 1: **Memory usage comparison (in GB) for a fixed compute overhead.** At 10% compute overhead over PyTorch, MONeT uses **2-3**× less memory than PyTorch. At the same overhead, MONeT can train models using 1.2-1.8× less memory than Checkmate.

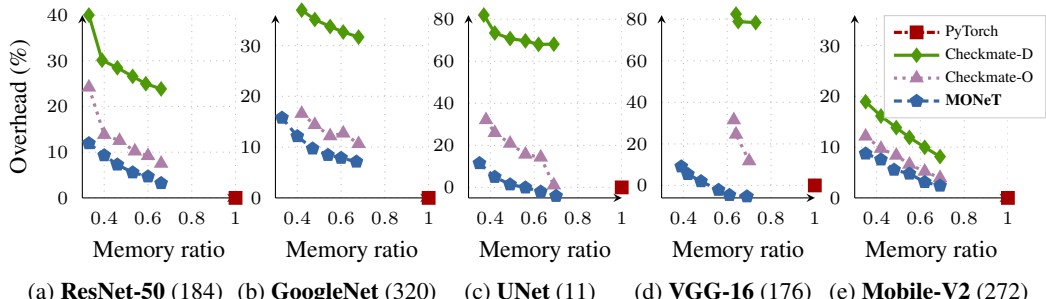

(a) **ResNet-50** (184)  (b) **GoogleNet** (320)  (c) **UNet** (11)  (d) **VGG-16** (176)  (e) **Mobile-V2** (272)

Figure 3: **Comparing MONeT with PyTorch and Checkmate.** MONeT reduces memory by $3\times$ compared to PyTorch, with 9-16% compute overhead. It achieves a better memory-compute trade-off than default Checkmate-D and conv-optimized Checkmate-O.

| | 5 GB | 6 GB | 7 GB | 8 GB | 9 GB | 10 GB |
|---|---|---|---|---|---|---|
| **ResNet-50** | | | | | | |
| Checkmate | - | 8.96 | 12.01 | 10.78 | 4.54 | 2.98 |
| MONeT-NoOp | 1.18 | 0.46 | 0.14 | 0.09 | 0.06 | 0.07 |
| MONeT | **7.24** | **3.84** | **0.73** | **0.70** | **0.31** | **0.11** |
| **GoogleNet** | | | | | | |
| Checkmate | - | 12.72 | 4.56 | 4.32 | 3.92 | 0.86 |
| MONeT-NoOp | 0.10 | 0.11 | 0.07 | 0.07 | 0.07 | 0.07 |
| MONeT | **3.53** | **0.47** | **0.54** | **0.31** | **0.25** | **0.24** |
| **VGG-16** | | | | | | |
| Checkmate | - | - | - | **0.002** | **0.002** | **0.001** |
| MONeT-NoOp | - | - | - | 0.001 | 0.000 | 0.000 |
| MONeT | - | 0.003 | 0.003 | 0.003 | 0.003 | 0.003 |

Table 2: **Solver time (in hours) to reach 5% close to optimal solution.** MONeT-NoOp reaches a 5% close-to-optimal solution $1.6\times$-$117\times$ faster than Checkmate. MONeT gets close to 5% of the optimal solution only in a few hours, and up-to $16\times$ faster than Checkmate for larger models.

MONeT and Checkmate schedules. The memory budgets range from 5 GB to 10 GB, or equivalently, $0.33\times$ to $0.70\times$ PyTorch memory consumption. Batch size for these models is mentioned in paranthesis. For all models, MONeT reduces memory usage by $3\times$ (0.33 memory ratio) as compared to baseline PyTorch with $9-16\%$ compute overhead. For the same memory budget, MONeT schedules are up-to 34% faster than Checkmate schedules. Note that we measure the empirical performance of the schedules running on GPUs instead of just providing a simulation of runtime and memory using the solver values; this is important since Checkmate does not consider workspace cost and overestimates its savings.

For networks with individual memory-intensive layers, like VGG-16, operator optimization becomes even more important for reducing memory; Checkmate can reduce memory for VGG-16 only up to 7 GB, whereas MONeT with its optimizations is able to run VGG-16 with 5.5 GB memory. The small runtime improvement of MONeT schedules over PyTorch for VGG-16 and UNet at higher memory budgets is mainly because of choosing faster convolution algorithms. MobileNet-V2 uses depthwise convolutions, and hence does not significantly benefit from joint convolution-optimization. As a result, the performance of MONeT and Checkmate is closer for MobileNet-V2. We provide additional results for MONeT on a memory-intensive model, 3D-UNet (Çiçek et al., 2016), in Appendix J, for which we observe a consistent memory reduction to $0.54\times$ of PyTorch memory with an overhead of 8.86%.

For our evaluation, we cap the solver time to 24 hours for both MONeT and Checkmate, and run the schedule thus obtained on our execution framework. At tighter memory budgets for non-linear models like ResNet-50 and GoogleNet, Checkmate is unable to find a feasible solution within a couple of hours. In contrast to Checkmate, MONeT finds the execution plans efficiently. For all the models and memory limits that we evaluate, MONeT reaches a 5% close-to-optimal solution within

| | VGG-16 (176) | | ResNet50 (184) | | GoogleNet (320) | | MobileNetV2 (256) | | UNet (11) | |
|---|---|---|---|---|---|---|---|---|---|---|
| | mem | overhead | mem | overhead | mem | overhead | mem | overhead | mem | overhead |
| Gist | 0.76 | 44.34 | 0.58 | 105.69 | 0.52 | 35.94 | 0.69 | 153.98 | 0.73 | 38.26 |
| **MONeT** | **0.39** | 9.11 | **0.33** | 11.94 | **0.33** | 15.77 | **0.34** | 8.80 | **0.35** | 11.51 |

Table 3: **Memory ratio and overhead (%) over PyTorch for Gist and MONeT.** MONET obtains 1.4×-2.1× higher memory savings over Gist across models. Number in parenthesis after model name shows the batch size.

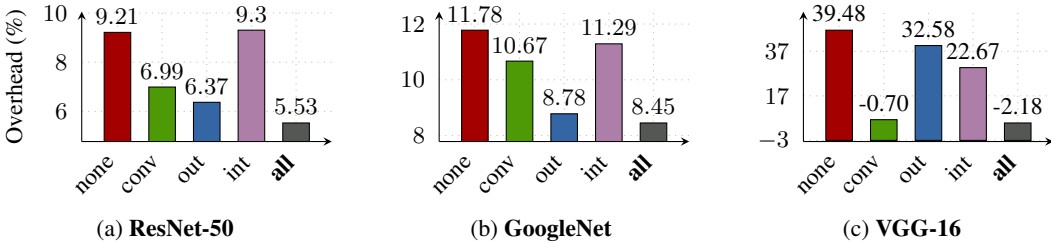

(a) **ResNet-50**  (b) **GoogleNet**  (c) **VGG-16**

Figure 4: **Ablation results for memory ratio 0.53.** Lowest compute overhead across models is seen only when all optimizations are jointly optimized.

a few hours or sometimes even minutes. Table 2 shows the time it takes for the solver to reach 5% close to the optimal solution, for Checkmate, MONET-NOOP (MONET with checkpointing enabled but operator-optimization disabled), and MONET. MONET-NOOP converges to a close-to-optimal solution 1.6×-117.4× faster than Checkmate. For larger models, MONET's solver converges to a close-to-optimal solution up to 27× faster than Checkmate. Note that running a solver is a one-time cost for a model - once a MONET schedule has been solved for, it can be used by everyone to train the model for different purposes with different batch sizes. The cost (typically seconds to hours) is tiny compared to the efforts and costs to develop a model for distribution in most cases. See Appendix H for more discussion regarding solver times, problem statistics, and full Table 2 data.

(b) *Operator optimizations*: Table 3 shows the comparison of MONET with Gist. While MONET can determine a range of memory-runtime tradeoffs, purely operator-optimization-based schemes like Gist only provide a single memory-runtime data point. For MONET, we show the memory-runtime data point with the most memory saving. MONET uses 1.4×-2.1× less memory than Gist for multiple architectures while maintaining full-precision. Overall, Gist provides impressive memory savings, but incurs a high computation cost to achieve the savings.

While we get similar memory saving results for reimplemented-Gist as Jain et al. (2018) for VGG-16, our compute overhead results are higher. This could be because of evaluations on different frameworks (PyTorch v/s CNTK) and different GPU models (Nvidia P100 v/s Nvidia Maxwell GTX Titan X). Gist uses dense to sparse conversion using `cusparseSdense2csr` in one of its techniques. For the first ReLU-Conv layer in VGG-16 (shape `(2207744,256)`), this function takes 144ms, which itself is 10% of the VGG-16 execution time. We see similar results for other networks. To ensure a fair comparison, we focus on the maximum memory savings obtained by MONET with Gist, while reporting the compute overhead for completeness.

**Ablation Experiments.** Fig. 4 shows additional ablation experiments. We show the % compute overhead over PyTorch on ResNet-50, GoogleNet, and VGG-16 for different types of MONET checkpointing schedules with a memory budget of 8 GB - with no operator optimizations enabled, with only one type of operator optimization enabled (conv-optimized, output-activated optimized, intermediate-activated optimized), and with all optimizations enabled. Schedules which do not jointly optimize convolution algorithms are run with greedily post-optimized convolution algorithm. Plots for other models look similar to that of ResNet-50 and GoogleNet. The only difference between 'none' and 'conv' is that convolution algorithms are jointly optimized in the latter. However, this fact leads to significant improvement in compute time for all cases. In fact, convolution algorithms have complex workspace memory - compute characteristics, reserving slightly more memory for convolution workspace while checkpointing can allow for a much faster convolution (see Appendix I). This makes it important to jointly optimize conv algorithms with checkpointing. Similarly, output-activated optimization also provides significant benefits over vanilla checkpointing, since it effectively reduces the number of recomputations required. For memory-intensive net-

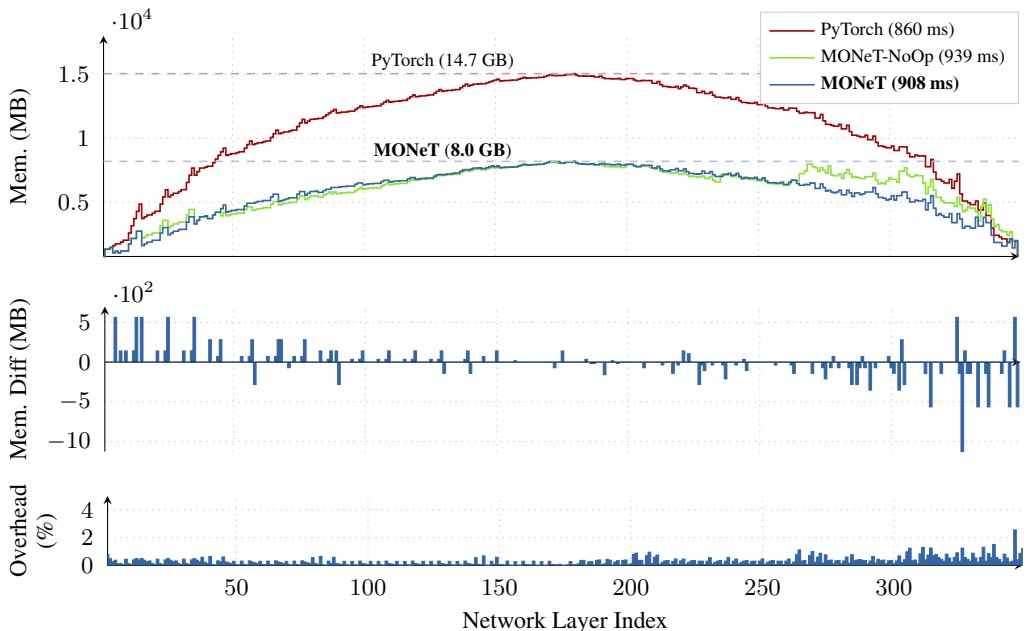

Figure 5: **Detailed case study on ResNet-50.** Top : memory usage along execution (forward and backward). Middle: memory saving of MONeT over PyTorch for each layer. Bottom: compute overhead of MONeT over PyTorch. MONeT saves memory in early layers to reduce peak memory. Most compute overhead happens at recomputation during backward (right-hand-side of the figure).

works, intermediate-activated optimization becomes more important. Jointly optimizing all strategies together gives the least computational overhead. See Appendix G for detailed ablation plots.

**Detailed Case Study.** The top graph of Fig. 5 shows memory usage while executing PyTorch, MONET without operator optimization, and MONET for ResNet-50 at batch size 184. As the training progresses along network layers represented on X-axis, PyTorch and both MONET schedules store forward-pass outputs, leading to an increasing memory footprint. MONET reaches peak memory of 8 GB, whereas PyTorch requires 14.7 GB. Stored forward outputs are freed up one after other as backward pass proceeds, leading to reduced usage of memory. According to the checkpointing schedule, MONET saves only a subset of the outputs stored by PyTorch, resulting in the memory saving shown in the middle graph for layer outputs that are not stored. The bottom graph shows the per-layer compute overhead of recomputation of MONET over PyTorch. For MONET, later layers which are backward operators result in a recomputation of the forward, and have higher overhead.

## 6 CONCLUSION

We present MONET, a system to automatically reduce memory requirements for training deep networks. MONET jointly optimizes local (operator-level) and global (graph-level) optimizations to yield a compute- and memory-efficient checkpointing schedule. MONET reduces memory usage by $3\times$ over PyTorch, with a $9 - 16\%$ compute overhead. It uses 1.2-1.8$\times$ less memory than the state-of-the-art automated checkpointing framework for the same computational cost. Our experimental results show that MONET leads to better memory-computation trade-offs compared to the state-of-the-art.

### ACKNOWLEDGMENTS

We would like to thank the anonymous reviewers for their feedback. Aashaka Shah and Vijay Chidambaram were partially supported by donations from VMware and Google. Chao-Yuan Wu was partially supported by a Facebook Fellowship. Jayashree Mohan was supported by a Microsoft Research Fellowship. The results presented in this paper were obtained using the Chameleon testbed supported by the National Science Foundation

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

# A  NOTATIONS

Table 4 gives a list of notations used in the paper along with explanations.

| Notation | Meaning |
|---|---|
| **Tensors** | |
| $x_i$ | Tensor which is the output of $\text{forward}_i$ in forward pass and during recomputation. |
| $y_k$ | Gradient tensor which is the output of $\text{backward}_i$ in the backward pass. |
| **Sets** | |
| $S_i^N$ | Set of stored tensors after $\text{forward}_i$ in forward pass. (N = num backward operators) |
| $L_i$ | Set of all parameters and forward tensors created till forward node $i$, required as computational dependencies for $\text{forward}_i$ and later forward passes. |
| $D_k$ | Set of forward pass tensors required as computational dependencies for $\text{backward}_k$. |
| $S^{k-1}$ | Set of stored forward pass tensors right before calling $\text{backward}_k$. |
| $\hat{L}_k$ | Set of gradient tensors created before backward node $k$, and required as computational dependencies for $\text{backward}_k$ and later backward passes. |
| $S_i^{k-1}$ | Set of stored tensors available to recomputation of $\text{forward}_i$ before computing $\text{backward}_k$. |
| $L_i^k$ | Set of all parameters and forward tensors created till forward node $i$, required as computational dependencies for $\text{forward}_i$ and later forward recomputations to be done before $\text{backward}_k$. |
| $\mathcal{I}_i$ | Set of implementations for operator $\text{forward}_i$. |
| $\hat{\mathcal{I}}_k$ | Set of implementations for operator $\text{backward}_k$. |
| **Solver variables** | |
| $s_i^N$ | Indicate if output of $\text{forward}_i$ is stored in memory in the forward pass. |
| $s_i^{k-1}$ | Indicate if output of $\text{forward}_i$ is stored in memory when computing $\text{backward}_k$. |
| $r_i^k$ | Indicate if $\text{forward}_i$ is recomputed before computing $\text{backward}_k$. |
| $\delta_{i,l}$ | Indicate if $\text{forward}_i$ uses implementation $l \in \mathcal{I}_i$ in the forward pass. |
| $\delta_{i,l}^k$ | Indicate if $\text{forward}_i$ uses implementation $l \in \mathcal{I}_i$ when recomputed before $\text{backward}_k$. |
| $\hat{\delta}_{k,l}$ | Indicate if $\text{backward}_k$ uses implementation $l \in \hat{\mathcal{I}}_k$. |
| **Memory formulations** | |
| $m_i$ | Peak memory of $\text{forward}_i$ in forward pass. |
| $\bar{m}_i^k$ | Peak memory of $\text{forward}_i$ when it is recomputed before $\text{backward}_k$. |
| $\hat{m}_k$ | Peak memory of $\text{backward}_k$. |
| **Operator costs** | |
| $c_i^l$ | Workspace memory of operator $\text{forward}_i$ executed using implementation $l \in \mathcal{I}_i$. |
| $\hat{c}_k^l$ | Workspace memory of operator $\text{backward}_k$ executed using implementation $l \in \hat{\mathcal{I}}_k$. |
| $\tau_i^l$ | Compute cost of operator $\text{forward}_i$ executed using implementation $l \in \mathcal{I}_i$. |
| $\hat{\tau}_k^l$ | Compute cost of operator $\text{backward}_k$ executed using implementation $l \in \hat{\mathcal{I}}_k$. |

Table 4: **Notations used in paper with explanations.** Notations with only $i$ in subscript/superscript generally relate to the forward pass, with only $k$ relate to the backward pass, and with both $i$ and $k$ relate to the recomputation phase.

# B  BOUNDS ON LOCAL MEMORY

In Section 3, we mentioned that local memory $L_i^k$ is dependent on solver variable $r_t^k$.

$$L_i^k = \Theta \cup \{x_j : j \in \mathcal{N}_t \text{ for any } t \geq i \text{ with } r_t^k = 1 \text{ and } j < i\}.$$

In order to remove this dependence, we can get an upper bound $\bar{L}_i^k$ on $L_i^k$ by assuming that all future tensors after $i$ will always be recomputed, that is, $r_t^k = 1, \forall t > i$, and

$$L_i^k \subseteq \bar{L}_i^k = \Theta \cup \{x_j : j \in \mathcal{N}_t \text{ for any } t \geq i \text{ and } j < i\}.$$

Our experiments also use this upper bound. It is possible to tighten the upper bound by noting that $r_t^k$ may be 1 only in the case when $t \leq k$. That is, forward node $t$ will not be recomputed before computing backward of node $k$ if node $t$ lies after node $k$. Thus, a tighter bound to $L_i^k$ follows

$$L_i^k \subseteq \dot{L}_i^k = \Theta \cup \{x_j : j \in \mathcal{N}_t \text{ for any } t \geq i \text{ and } t \leq k \text{ and } j < i\} \subseteq \bar{L}_i^k.$$

## C  DETAILED CONSTRAINTS

### C.1  EXPANDED BACKWARD PASS MEMORY CONSTRAINTS

Sec. 4 formulates backward peak memory $\hat{m}_k$ and recomputation peak memory $\bar{m}_i^k$ as sum of memory of a set of tensors. We expand the memory formulation and represent it in the terms of optimization varaible here:

$$\hat{m}_k = \sum_l \hat{c}_k^l \hat{\delta}_{k,l} + |y_k| + |\hat{L}_k| + \sum_l \hat{\delta}_{k,l}.|D_k^l \cup S^{k-1}|$$

$$= \sum_l \hat{c}_k^l \hat{\delta}_{k,l} + |y_k| + \sum_{y_l \in \hat{L}_k} |y_l| + \sum_l \sum_{x_i \in D_k^l} \hat{\delta}_{k,l}|x_i| + \sum_l \sum_{x_i \notin D_k^l} \underbrace{\hat{\delta}_{k,l} s_i^{k-1}}_{\sigma_{k,l,s}} |x_i|, \quad (10)$$

$$\bar{m}_i^k = c_i + |x_i| + |\hat{L}_k| + \sum_l \hat{\delta}_{k,l}.|S_i^{k-1} \cup \bar{L}_i^k \cup D_k^l|$$

$$= c_i + |x_i| + |\hat{L}_k| + \sum_l \sum_{\substack{j<i: \\ x_j \notin \bar{L}_i^k \cup D_k^l}} \hat{\delta}_{k,l} s_j^{k-1} |x_j| + \sum_l \sum_{\substack{j<i: \\ x_j \in \bar{L}_i^k \cup D_k^l}} \hat{\delta}_{k,l}|x_j| + \sum_{j>i} s_j^k |x_j|. \quad (11)$$

### C.2  COMPLETE MEMORY CONSTRAINTS

In this section, we present the complete memory constraints which we use for MONET optimization. These constraints include the recomputation variable $r_i^k$, which was excluded from the main text to make understanding simpler. As discussed in Sec. 3, the peak memory of a $\mathrm{forward}_i$ recomputation before computing $\mathrm{backward}_k$ is denoted by $\tilde{m}_i^k$. This represents the recomputation memory (renamed to $m_{Ri}^k$) when $\mathrm{forward}_i$ is actually recomputed, that is, $r_i^k = 1$. When this is not true, the peak memory ($\tilde{m}_{Si}^k$) only depends on stored checkpoints $S_i^{k-1}$, checkpoint dependencies for $D_k$, and gradient tensors $\hat{L}_k$. Thus,

$$\tilde{m}_{Ri}^k = c_i + |x_i| + |\hat{L}_k| + |S_i^{k-1} \cup L_i^k \cup D_k|$$

$$= r_i^k c_i + r_i^k |x_i| + |\hat{L}_k| + \sum_{j<i:x_j \notin L_i^k \cup D_k} s_j^{k-1} |x_j| + \sum_{j<i:x_j \in L_i^k} r_i^k |x_j| + \sum_{j<i:x_j \in D_k - L_i^k} |x_j| + \sum_{j>i} s_j^k |x_j|. \quad (12)$$

$$\tilde{m}_{Si}^k = |\hat{L}_k| + |S_i^{k-1} \cup D_k|$$

$$= |\hat{L}_k| + \sum_{j \leq i:x_j \notin D_k} s_j^{k-1} |x_j| + \sum_{j \leq i:x_j \in D_k} |x_j| + \sum_{j>i} s_j^k |x_j|. \quad (13)$$

Local memory $L_k$ can be bounded by $\bar{L}_k$, which gives us $\bar{m}_{Ri}^k$. To add forward operator optimizations to $\bar{m}_{Ri}^k$, we recall the trade-off between workspace memory and compute time. We replace the workspace memory contributor $r_i^k c_i$ in equation 12 with $\sum_l \delta_{i,l}^k c_i^l$.

The complete memory constraints are:

$$m_i \leq M \quad \forall_i \quad \text{and} \quad \hat{m}_k \leq M \quad \forall_k \quad \text{and} \quad \bar{m}_{Ri}^k \leq M \quad \forall_{k,i} \quad \text{and} \quad \tilde{m}_{Si}^k \leq M \quad \forall_{k,i} \quad (14)$$

### C.3  IN-PLACE CONSTRAINTS

We show how to represent the decision of computing an operator using an in-place or out-of-place implementation. If an operator like ReLU uses an in-place implementation, its input tensor is overwritten with its output. In this case, its input tensor cannot be stored or used as input to a computation in this stage. This needs to be reflected in our constraints. We introduce two new binary variables to model in-place computations: $q_i^k$ represents if $\mathrm{forward}_i$ is recomputed in-place when computing $\mathrm{backward}_k$. $p_i^k$ represents that the output of $\mathrm{forward}_i$ has been computed and will not be overwritten by any other forward node recomputations in this stage. If $q_i^k$ is true, then $p_j^k$ will be false else $p_j^k$ will be the same as $r_j^k$, where $j \in \mathcal{N}_i$. Further, $s_j^{k-1}$ will also be false if $q_i^k$ is true. This can be written in the form of boolean constraints as follows:

$$p_j^k \geq r_j^k - 2q_i^k \quad \text{and} \quad p_j^k \leq 2 - 2q_i^k \quad \text{and} \quad s_k^{k-1} \leq 2 - 2q_i^k. \quad (15)$$

The checkpointing constraint 7 changes, with $p_j^k$ replacing $r_j^k$ on the RHS. Further, $q_i^k$ (or $p_j^k$) can only be true if $\text{forward}_i$ (or $\text{forward}_j$) is actually recomputed prior to computing backward node k. Thus,

$$p_j^k \leq r_j^k \qquad \text{and} \qquad q_i^k \leq r_i^k. \tag{16}$$

### C.4 Constraint Linearization

The memory constraints we introduce in Section 4 contain quadratic terms in the form of $x_i \cdot x_j$, with $x_i, x_j \in \{0, 1\}$. The quadratic terms cannot directly be incorporated into an integer program. However, we can linearize these terms by replacing each quadratic term $x_i \cdot x_j$ by an auxiliary variable $\alpha_{i,j} \in \{0, 1\}$ and introducing additional linear constraints $\alpha_{i,j} \geq x_i + x_j - 1$, $\alpha_{i,j} \leq x_i$, and $\alpha_{i,j} \leq x_j$. After this substitution for all quadratic terms, all constraints in MONET are linear.

## D Implementation

We develop MONET in the PyTorch (v1.5.1) framework. We use PyTorch's default Autograd package for backward implementation of elementary functions when the autograd implementation is stateless. In all other cases, we implement custom forward and backward functions leveraging PyTorch ATen library functions to flexibly support multiple operators and execution schedules. Each backward operator implementation is annotated with its computational dependencies, which is generally the input or the output of its corresponding forward operator. Certain backward operators implementations may have dependencies on intermediate activations generated in the forward pass. For example, an intermediate-activated ReLU backward uses an encoded bitmask representing the sign of forward operator's input. We annotate this as an intermediate storage node and add it to our optimization problem, with a strict recomputation dependency of the interemediate storage node on its creator node. Our operator optimizations select from different backward operator implementations, convolution algorithms, in-place operators etc. We split the convolution backward operator into two - a parameter-gradient operator followed by an input-gradient operator. Since the input-gradient operator does not have any computational dependency on the forward pass, we can agressively free the forward input tensor right after the parameter-gradient is computed. We also reuse BatchNorm statistics in case of their recomputation. For our experiments, we limit ourselves to full precision training as quantization or lower precision computations introduce additional noise into SGD and change its convergence properties. We solve the joint optimization problem using the CVXPY (Diamond & Boyd, 2016; Agrawal et al., 2018) solver with Gurobi (2014) backend.

**MONET workflow.** We obtain the forward pass dependencies in MONET by JIT tracing a model to obtain its graph. We profile each layer for workspace memory and compute cost, and obtain memory usage of the tensors from their shape and type. Note that the workspace memory for many convolution operators in VGG-16 is greater than 2GB, making it an important factor to model. Unlike prior approaches like Checkmate, we account for this workspace memory in our optimization problem, bringing the memory model very close to actual memory allocation. We phrase a boolean integer programming problem using the generated graph and the profiled compute cost and workspace memory and solve it using the CVXPY (Diamond & Boyd, 2016; Agrawal et al., 2018) modeling language and GUROBI (Gurobi, 2014) solver. The solution is used to generate a schedule that can be run by the MONET scheduler.

**Operator optimizations.** We divide operator optimizations according to the different type of implementations they select from. (1) *Output-activated*: Backward calculation of operators like ReLU and BatchNorm can have computational dependency either on on their forward node's inputs or outputs. (2) *Intermediate-activated*: Backward of ReLU has computational dependency on a 1-bit encoding of the sign of its forward node's input. Backward of MaxPool is calculated using an intermediate 8-bit output-shaped tensor which contains the kernel-index of the maximum element. (3) *Convolution algorithms*: We choose from 8 forward and 6 backward cuDNN convolution algorithms. (4) *Inplace operations*: The solver can choose to do inplace computation for operators like ReLU forward. We discuss constraints for in-place operator selection in C.3. All MONET experiments enable in-place operation selection.

## E    BASELINE IMPLEMENTATIONS

**Checkmate implementation.** We reimplement Checkmate (Jain et al., 2019) in PyTorch for our comparisons. We use the Checkmate solver as-is to obtain Checkmate schedules. Since Checkmate does not provide an execution engine for PyTorch, we run the generated Checkmate schedules on our own execution framework. Our inference engine uses the same operator implementations for Checkmate and MONeT. We have released our Checkmate implementation with the MONeT code.

**Gist implementation.** Gist (Jain et al., 2018) is an operator-based memory-efficient scheme for training DNNs. It encodes stashed forward tensors into smaller tensors which require less memory. Jain et al. (2018) evaluate Gist using CNTK on an Nvidia Maxwell GTX Titan X GPU. Since we implement MONeT in PyTorch and have access to an Nvidia P100 GPU, a direct comparison with the numbers in the Gist paper is not possible. As an official Gist implementation is not available, we reimplement it on PyTorch and evaluate its execution using MONeT 's execution framework.

We implement all Gist optimizations — Binarize (intermediate encodings for ReLU-Pool layers), Sparse Storage Dense Compute (compress and store sparse convolution inputs in ReLU-Conv layers as sparse storage), Delayed Precision Reduction (storing stashed non-compressed tensors in FP-16, but computing in FP-32), and Inplace (performing ReLU operators in-place wherever possible) over MONeT's execution framework. In Gist, the Sparse Storage Dense Compute (SSDC) technique creates a sparse storage tensor in the Compressed Sparse Row (CSR) representation using the Nvidia cuSPARSE library. The dense storage is reshaped into a 256-sized column tensor before storing it in a sparse format, allowing the column index of CSR representation to be saved in 8 bits instead of using 32 bits (termed Narrow Value Optimization in the paper). We also implement SSDC using Nvidia's cuSPARSE library (function `cusparseSdense2csr`) with CUDA Toolkit version 10.1 using PyTorch's C++ extensions.

In their paper, Jain et al. (2018) use the most memory-efficient convolution algorithms in Gist and compare its memory saving against a baseline which also chooses the most memory-efficient convolution algorithm. Using memory-efficient convolution algorithms, our Gist reimplementation can train VGG-16 with $0.55\times$ of the PyTorch-required memory ($1.81\times$ memory footprint), which is close to the data presented by Jain et al. (2018). However, it is 59% slower than when convolution selection is enabled, in which case it can train using $0.76\times$ of the PyTorch-required memory. Since implementing Gist using memory-efficient convolutions is not optimal in terms of compute time, we implement Gist to use PyTorch's convolution selection algorithm. For all models other than VGG-16 and UNet, we see similar memory savings for Gist with memory-efficient convolutions and with convolution-selection enabled. We have released our Gist implementation with the MONeT code.

## F    ON OPERATOR SELECTION FOR CHECKMATE

In this section, we briefly explain the difficulties of including operator selection directly into check-mate (Jain et al., 2019). We will refer directly to notation and equations in the checkmate paper (arxiv v3; 14 May 2020). The most direct way to incorporate operator selection into checkmate is to introduce an auxiliary variable $R_{t,i}^v \in \{0, 1\}$ that refers to re-computing layer $i$ at time $t$ using implementation $v$. Most constraints in equation 1 could stay the same, given $R_{t,i} = \sum_v R_{t,i}^v$, and loss (1a) $\sum_t \sum_i \sum_v R_{t,i}^v C_i^v$. Some of our operators produce a different kind of checkpoint (e.g. binary activated ReLUs), which could be handled in check-mate by splitting $S_{t,i}^v$. The main issues in checkmate arise in the memory modeling and its relaxations (eq 4,5,7). The memory consumed by a specific checkpoint may depend on the operator implementation: DEPS[k] and USERS[i] both depend on the operator implementation (output activated, input activated, ...). In short, the checkmate computation graph is dynamic and depends on operator implementations. The most direct way to address this is to $\mathrm{mem\_freed}_t(v_k) = \sum_v R_{t,i}^v \mathrm{mem\_freed}_t(v_k)$ in a implementation-dependent way $\mathrm{mem\_freed}_t^v(v_k)$, and select the right version dependent on the operator used. Likewise, we need to extend $\mathrm{FREE}_{i,t,k}^v$ to account for different operator implementations in $R_{t,k}^v$. Likewise the product in equation (5) will now go over all implementations $R_{i,j}^v$ using different USERS sets. This leads to a linear blowup in the number of constraints, and number of auxiliary variables, leading to an at least quadratic expansion on computational costs. Furthermore, $\mathrm{mem\_freed}_t(v_k) = \sum_v R_{t,i}^v \mathrm{mem\_freed}_t(v_k)$ is a quadratic constrain that further needs to be resolved using additional auxiliary variables. Given that Checkmate already pushes the limits of

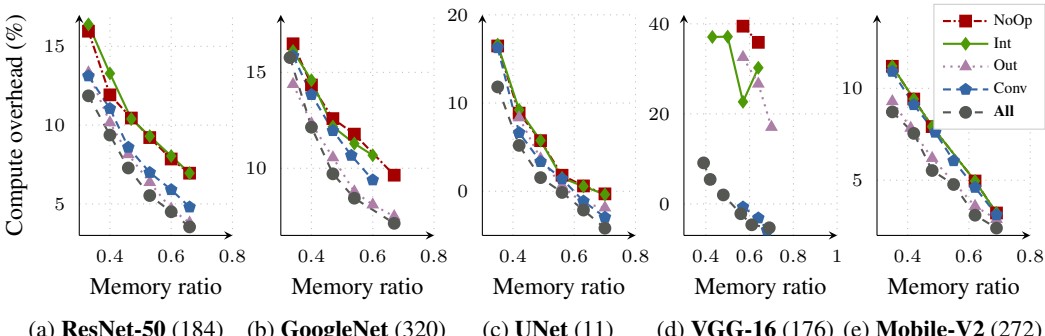

(a) **ResNet-50** (184)  (b) **GoogleNet** (320)  (c) **UNet** (11)  (d) **VGG-16** (176) (e) **Mobile-V2** (272)

Figure 6: **Ablation results on ResNet-50, GoogleNet, UNet, VGG-16, MobileNet-V2.**

current solvers, it is unlikely able to handle this explosion in constraints and variables, without significant modifications. MONET in the other hand represents the compute-graph more compactly and efficiently integrates different operator implementations.

## G  DETAILED ABLATIONS

Fig. 6 shows a detailed plot of our ablation experiments comparing the compute overhead of variants of MONET across a range of memory limits. Y-axis shows the compute overhead over PyTorch and X-axis shows the memory ratio to a PyTorch model. All variants which are not conv-optimized are greedily post-optimized to use the fastest convolution. We see that MONET with no operator optimization (NoOp) is generally slower than the other variants for all models and memory limits. Convolution and output-activated optimizations are both important in reducing compute overhead. MobileNet-V2 uses depthwise separable convolutions, and hence does not significantly benefit from convolution-optimization. Further, MobileNet-V2 has `hardtanh` operators instead of ReLU operators, for which we have not implemented intermediate-activated backward optimization. Interemediate-activated optimizations provide memory savings in memory-intensive models, allowing models like VGG-16 to reach memory savings which are not attainable by other optimizations. All optimizations together result in the least compute overhead for any model or memory limit.

## H  SOLVER TIME AND ILP STATISTICS

**Solver runtime.** MONET's solver runtimes vary for different models and different memory limits. We evaluate schedules obtained using solver times set to a maximum of 24 hours. For moderate memory limits, both MONeT and Checkmate achieve an optimal solution before 24 hours. For tighter memory limits, the solution obtained by MONeT and Checkmate may not be most optimal. For multiple models and memory limits, Table 2 in Sec. 5 shows the time it takes for the solver to reach 5% close to the optimal solution for Checkmate, MONET-NOOP (MONET with checkpointing enabled but operator-optimization disabled), and MONET. We add the data for MobileNet-V2 and UNet in Table 5 which also follow a similar pattern. We also provide Table 6 which shows the time taken by the solver to reach 2% close to the optimal solution. We note that it has similar behavior as the time taken the solver to reach 5% close to the optimal solution. MONET-NOOP converges to 2% close-to-optimal solution 1.3×-139× faster than Checkmate. For larger models, MONeT's solver converges to a 2% close-to-optimal solution up to 16× faster than Checkmate. At tighter memory limits for MobileNet-V2, the Checkmate solver reaches 2% close-to-optimal solution faster than MONET, but is still much slower than MONET-NOOP.

**ILP statistics.** For different models, Table 7 shows the solver statistics after presolving for the problem formulated by Checkmate, MONET-NOOP and MONET for a 10 GB memory limit. It shows the number of forward operators in the model and the number of constraints and variables for each solver. MONET-NOOP, which is MONET with only checkpointing enabled and without using operator optimization, has on average 50% fewer constraints and 67% fewer variables than Checkmate. Jointly-optimized MONET has a slightly larger number of constraints, and on average

| | 5 GB | 6 GB | 7 GB | 8 GB | 9 GB | 10 GB |
|---|---|---|---|---|---|---|
| **ResNet-50** | | | | | | |
| Checkmate | - | 8.96 | 12.01 | 10.78 | 4.54 | 2.98 |
| MONeT-NoOp | 1.18 | 0.46 | 0.14 | 0.09 | 0.06 | 0.07 |
| MONeT | **7.24** | **3.84** | **0.73** | **0.70** | **0.31** | **0.11** |
| **GoogleNet** | | | | | | |
| Checkmate | - | 12.72 | 4.56 | 4.32 | 3.92 | 0.86 |
| MONeT-NoOp | 0.10 | 0.11 | 0.07 | 0.07 | 0.07 | 0.07 |
| MONeT | **3.53** | **0.47** | **0.54** | **0.31** | **0.25** | **0.24** |
| **MobileNet-V2** | | | | | | |
| Checkmate | 2.16 | 2.88 | 1.16 | 0.29 | 0.34 | 0.14 |
| MONeT-NoOp | 0.11 | 0.04 | 0.02 | 0.02 | 0.04 | 0.08 |
| MONeT | **0.37** | **0.28** | **0.52** | **0.05** | **0.06** | **0.03** |
| **UNet** | | | | | | |
| Checkmate | **0.149** | **0.031** | **0.022** | **0.020** | **0.010** | 0.009 |
| MONeT-NoOp | 0.048 | 0.002 | 0.002 | 0.002 | 0.002 | 0.002 |
| MONeT | 0.363 | 0.064 | 0.028 | 0.027 | 0.024 | **0.006** |
| **VGG-16** | | | | | | |
| Checkmate | - | - | - | **0.002** | **0.002** | **0.001** |
| MONeT-NoOp | - | - | - | 0.001 | 0.000 | 0.000 |
| MONeT | - | **0.003** | **0.003** | 0.003 | 0.003 | 0.003 |

Table 5: **Solver time (in hours) to reach 5% close to optimal solution.** MONeT-NoOp reaches a 5% close-to-optimal solution $1.6\times$-$117\times$ faster than Checkmate. MONeT gets close to 5% of the optimal solution only in a few hours, and up-to $16\times$ faster than Checkmate for larger models.

| | 5 GB | 6 GB | 7 GB | 8 GB | 9 GB | 10 GB |
|---|---|---|---|---|---|---|
| **ResNet-50** | | | | | | |
| Checkmate | - | **16.44** | 13.43 | 11.91 | 5.74 | 3.81 |
| MONeT-NoOp | - | 2.06 | 1.28 | 0.16 | 0.08 | 0.07 |
| MONeT | - | - | **12.64** | **3.00** | **3.60** | **0.62** |
| **GoogleNet** | | | | | | |
| Checkmate | - | 15.08 | **4.93** | 5.04 | 3.92 | 0.90 |
| MONeT-NoOp | 0.10 | 0.11 | 0.07 | 0.07 | 0.07 | 0.07 |
| MONeT | - | **5.47** | 5.34 | **0.31** | **0.25** | **0.24** |
| **MobileNet-V2** | | | | | | |
| Checkmate | **2.16** | **2.88** | **1.16** | 0.29 | 0.34 | 0.14 |
| MONeT-NoOp | 0.43 | 0.37 | 0.02 | 0.02 | 0.10 | 0.09 |
| MONeT | 9.49 | 5.33 | 1.53 | **0.14** | **0.18** | **0.05** |
| **UNet** | | | | | | |
| Checkmate | **0.243** | **0.031** | **0.027** | **0.021** | **0.011** | **0.009** |
| MONeT-NoOp | 0.181 | 0.003 | 0.003 | 0.002 | 0.002 | 0.002 |
| MONeT | 5.001 | 0.204 | 0.164 | 0.069 | 0.083 | 0.027 |
| **VGG-16** | | | | | | |
| Checkmate | - | - | - | **0.003** | **0.002** | **0.001** |
| MONeT-NoOp | - | - | - | 0.001 | 0.000 | 0.000 |
| MONeT | - | **0.004** | **0.006** | 0.004 | 0.003 | 0.003 |

Table 6: **Solver time (in hours) to reach 2% close to optimal solution.** MONeT-NoOp reaches a 2% close-to-optimal solution $1.3\times$-$139\times$ faster than Checkmate. MONeT reaches a 2% close-to-optimal solution within few hours in most cases, and up to $27\times$ faster than Checkmate for larger models.

40% fewer variables than Checkmate. MONeT's formulation is more efficient, and might be the reason that it reaches a good solution faster than Checkmate.

| | Fwd ops | Checkmate | | MONeT-NoOp | | MONeT | |
|---|---|---|---|---|---|---|---|
| | | Constraints | Variables | Constraints | Variables | Constraints | Variables |
| GoogleNet | 215 | 719,327 | 519,252 | 221,630 | 104,673 | 781,747 | 362,640 |
| ResNet-50 | 175 | 473,592 | 344,659 | 344,652 | 167,238 | 487,842 | 229,431 |
| Mobilen.V2 | 153 | 337,316 | 247,033 | 153,828 | 74,579 | 241,478 | 115,047 |
| UNet | 67 | 65,715 | 48,744 | 32,273 | 15,982 | 73,624 | 36,548 |
| VGG-16 | 40 | 25,334 | 18,968 | 12,772 | 6,306 | 21,918 | 11,234 |

Table 7: **ILP statistics for Checkmate, MONeT-NoOp, and MONeT.** MONeT-NoOp has on average 50% fewer constraints and 67% fewer variables than Checkmate. MONeT has slightly larger number of constraints, on average 40% fewer variables than Checkmate.

## I  CONVOLUTION ALGORITHMS

Fig. 7 shows the complex workspace memory-compute trade-off for different convolution algorithms. The memory used is not always inversely proportional to the compute requirement. Jointly optimizing convolution algorithms enables MONET to make the best decisions for which convolution algorithm to select.

## J  APPLICABILITY TO MEMORY-INTENSIVE MODELS

To further show MONET's applicability to memory-intensive models, we evaluate it on 3D-UNet (Çiçek et al., 2016), a fully-convolutional model for volumetric images. Fig. 8 presents the runtime-memory trade-off for MONET on 3D-UNet. We used a commonly used 3D-UNet implementation (Wolny, 2019; Wolny et al., 2020) with training configuration similar to `3DUnet_confocal_boundary` provided in the repository and a batch size of 22, which just fits on a 16 GB P100 GPU. MONET reduces memory usage to $0.54\times$ of PyTorch, while incurring 8.86% overhead in compute time. At a memory ratio of 0.81, MONET incurs almost no computational overhead, because it makes use of operator optimizations and is able to bring down the recomputation cost to zero.

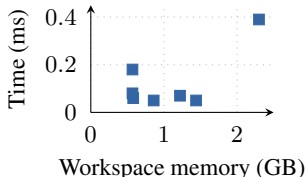

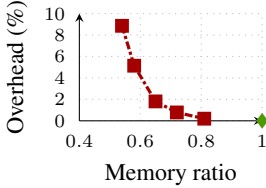

Figure 7: **Memory vs. compute** for 7 convolution algorithms with $256\times64\times56\times56$ input, $3\times3$ kernel, 64 output channels.

Figure 8: **Runtime-memory trade-off curve** for 3D-UNet using MONET. The green point denotes the PyTorch baseline.

