# OpenReview forum: "Memory Optimization for Deep Networks"
_ICLR.cc/2021/Conference — ICLR 2021 Spotlight_

### Official Review · AnonReviewer3 · 2020-10-28
**Good paper; please answer the questions posed**

**Rating:** 7
**Confidence:** 4

**Review:**

## Summary:

The paper provides a framework (MoNET) to perform automatic memory optimization targeting deep neural networks. Their technique jointly optimizes the checkpointing schedule and the choice of individual ops to find the implementation with the least possible runtime overhead under certain memory constraints. They formulate the problem as a ILP problem where the objective is to minimize runtime subject to strict memory constraints. They show that the solutions achieve 3x reduction in memory footprint compared to a pytorch implementation with minimal runtime overhead.

## Strengths:
* Making joint decisions on checkpointing and on the implementation of individual ops. This enables the authors to make both global and local decisions and to exploit the synergies in both.
* Experiments on multiple NN models to show the impact of their technique.
* Easy to read formulation of the ILP problem.

## Weaknesses and Questions for authors:
* Solution times and scalability
The authors do not explicitly mention the solution times for their ILP formulation. Also, it is unclear to me how the ‘checkmate’ framework can be slower compared to MoNET (as authors claim in page 7), since checkmate is only solving for the optimal checkpointing schedule.  I am also curious to know the scalability of the system for larger problems, specially since the authors are doing a joint optimization. Statistics about ILP problem sizes would be helpful.
* Questions about the formulation
  * Why do you need to compute $r_i^k$ for $i > k$? The gradient computations of op $k$ only depends on ops with numbers > $k$ (Algorithm 2)
  * How do you get the computational efficiency of ops? If fusion of ops happens inside the compiler, then adding up the computational efficiency of individual ops may not realistically model the execution behavior.
  * The formulation does not take into account possible compiler optimizations such as fusion, tiling etc. for both memory consumption and runtime calculations.
* Questions about experiments
  * It’s unclear what implementations of ops are used in the PyTorch implementation? Are these default ops or did you do a post-optimization similar to what you did for checkmate?
  * What if you set memory ratio = 1 and compare it with the pytorch implementation, what is the overhead?

---

> ### Author Response · Authors · 2020-11-25
> **Response to Reviewer 3**
>
> We thank the reviewer for the insightful questions and analysis.
> We provide our responses below:
>
> ### Solution times and scalability
>
> **[Solution times for the ILP formulation]**
> We evaluate schedules obtained using solution times set to a maximum of 24 hours.
> We have added Table 3 and Table 4 in Appendix H with detailed discussion of times to obtain close-to-optimal solutions for multiple models with multiple memory budgets.
> For all the 5 architectures and 30 memory limits that we tried, MONeT reached a 5% close-to-optimal solution within a few hours or sometimes minutes.
> We additionally include a comparison to Checkmate in the table; for larger models, MONeT's solver converged to a 5% close-to-optimal solution up to 27x faster, and to a 2% close-to-optimal solution up to 16x faster than Checkmate.
>
> ---
>
> **[Why is Checkmate's solver time slower than MONeT?]**
> MONeT and Checkmate differ in their translation of the computation graph into constraints.
> MONeT-NoOp, which is MONeT with only checkpointing enabled and without using operator optimization, has on average 50% fewer constraints and 67% fewer variables than Checkmate.
> Jointly-optimized MONeT has a slightly larger number of constraints and on average 40% fewer variables than Checkmate.
> MONeT’s formulation is more efficient, and might be the reason that it reaches a good solution faster than Checkmate.
>
> ---
>
> **[Scalability of the system for larger problems & ILP Statistics]**
> We have added data about solver times and ILP statistics for Checkmate, MONeT-NoOp (joint operator-optimization disabled), and MONeT in Appendix H. Three of the models we investigate are relatively large, containing more than 150 operators.
>         MONeT converges to a 5% close-to-optimal solution for the problems within a few hours.
>
> We would also like to point out that the 24-hours of solver time is a one-time effort - once a MONeT schedule has been solved, it can be used by all users for different purposes and can be extended to multiple batch sizes.
> This cost is usually tiny compared with the costs or time required to develop a model for distribution.
>
>
> ### Questions about the formulation
>
> **[Gradient computations of op k only depends on ops > k (Alg 2)]**
> This is a good point.
>     We have added a small discussion about it in Appendix B where we talk about tightening the upper bound on local memory.
>     In practice, we notice that while this reduces the ILP problem size, the solver time remains similar as before.
>
> ---
>
>
> **[How to get computational efficiency of ops? If fusion happens in compiler, then sum of individual ops may not realistically model the execution behavior.]**
> We obtain the computation graph of the model by JIT tracing the model and run the graph through optimizations like constant propagation and dead code elimination.
>     The graph nodes obtained are low-level pytorch primitives (aten::relu_, aten::_convolution, etc), for which we implement custom forward and backward functions leveraging the PyTorch ATen library functions.
>     We get the computational efficiency by profiling these ops.
>
> PyTorch by default does not fuse the low-level primitives for training. We too do not benchmark or execute with operator fusion currently in MONeT, but it is a great idea for future work.
>
> ---
>
>
> **[The formulation does not take into account compiler optimizations such as fusion, tiling etc.]**
> That’s correct.
>     We currently do not profile runtimes with compiler optimizations such as fusion and tiling enabled, or use them in our actual execution.
>     As mentioned above, we think it is a very good idea to consider compiler optimizations with MONeT.
>
>
> ### Questions about experiments
> **[What implementations of ops are used in the PyTorch implementation? Are these default ops or post-optimization similar to checkmate?]**
> We use PyTorch’s default ops.
>     We enable cuDNN benchmarking for PyTorch, which, much like the greedy post-optimization of convolutions in Checkmate, internally allows PyTorch to select the fastest convolution algorithm.
>
> ---
>
>
> **[What if memory ratio = 1 and compare it with the PyTorch implementation?]**
> On setting memory ratio = 1 and comparing it with the PyTorch implementation, we see a -3% to 7% overhead for MONeT.
>     Runtimes for ResNet-50 and MobileNet-V2 are very similar to PyTorch (0.5% and -0.3% overhead of MONeT over PyTorch respectively).
>     GoogleNet and UNet are slightly slower than PyTorch (by 7% and 4.3% respectively) owing to the presence of many MaxPooling layers, for which our implementation is slightly slower.
>     We observe that MONeT's VGG-16 selects faster convolution algorithms than PyTorch in a couple of cases, resulting in a slight net performance gain (overhead of -3%).

---

### Official Review · AnonReviewer1 · 2020-10-28
**Intuitive approach and framework to push the state-of-the-art in memory-constrained deep learning**

**Rating:** 7
**Confidence:** 4

**Review:**

The authors present MONeT, an automatic approach to jointly optimize operator cost and checkpoint scheduling for deep learning on a fixed memory budget.  The paper thoroughly defines the problem, relevant previous work, and the MONeT framework.  Given a fixed GPU memory budget, MONeT solves an integer program in order to jointly minimize the computational overhead of checkpointing with various operator implementation.  This approach is intuitive, as previous approaches, such as the recently proposed CheckMate, only optimize the checkpoint schedule.  The derived integer program is also a nontrivial extension of previous work.  With MONeT implemented in PyTorch, a large number of empirical results are presented, which show the superiority of MONeT compared to CheckMate, and show memory savings (versus impressively slight overhead) compared to PyTorch.

The paper is well written, the description of computational cost and the derivation of the integer program were interesting, and results are very compelling (and easy to understand).  One area where the paper could be improved is on the notation used throughout Section 4 (see below for suggestions), which was difficult to follow due to the density of the section, the large number of different variables/variants of variables described, and some implicit definitions.  There are also a few small details and discussions which seem warranted, but, overall, I enjoyed reading this paper.

Specific comments:

-"n Checkmate, changes in operator implementation induce a different computation graph" <- While this is
technically true, only the cost associated with that operator changes,
yes?  In this way, Checkmate could run multiple passes over the
computed static graph with different operator costs, but this approach
would require an expoential (in the number of operators with varying
costs) number of evaluations.

-"We reimplement Checkmate in PyTorch" <-This is non-trivial, so please include some re-implementation details.

-Please include the version of PyTorch which was forked for the Monet  and Checkmate implementations in Section 5.

-As CheckMate currently only supports tensorflow, it would be very helpful if the authors could also release the source for CheckMate in PyTorch when the source for MONeT is released.

-Could you comment on the solver runtime needed to solve the integer
program in monet, in contrast to the solver runtime for the MILP in
checkmate?

-How do open-source solvers compare to the runtime for the commerical
Gurobi solver for the integer programs solved in monet?

-"We measure the empirical performance of the checkpointed schedules
running on GPUs instead of just providing the solver values; this is
important since Checkmate doesn’t consider workspace cost and
overestimates its savings... Hence,
we show the results with solver running for 1 day for both MON E T and
Checkmate. In contrast, MON E T finds the execution plans efficiently,
its 1-hr solution already close to the 1-day solution with a small
difference of 1-2%."  <- This paragraph of worded somewhat vaguely; is
this to say 24 hours were included in execution time?

-For VGG-16 in the ablation study in Figure 4, PyTorch exceeds
device-memory for this dataset, yes? Is this the reason why monet
achieves negative overhead; i.e., faster execution time than PyTorch
itself?

Comments/questions regarding notation:

-What is variable r in the schedule (s,r)?  It *seems* like r is the
 indicator function for activations which require recomputation.  Is
 that correct?  If so, please (please) state this explicitly in the
 paper.  If not, please define r.

-A supplementary table for variables would be helpful.  It took some
 time to find a definition for $y$ in Equation 2.  While I eventually
 found one in Algorithm 2 (please define this explicitly inline,
 preceding Equation 2), a table would have made this much easier,
 especially considering how dense Section 3 is.

-By the time the reader gets to Section 4, the table of variables
 becomes mandatory (there are different values of L and S with various
 superscripts, subscripts, and hats, it is very difficult to recall
 which is which and to look back in the dense text for their
 definitions).

-Also, if possible, please standardize notation by the three
 categories:
(a) peak
 forward pass memory consumption
(b) peak backward pass memory consumption
(c) peak recomputation memory consumption
so that a reader can ascertain what collection Ls, Ds, and Ss are
being referred to.  I understand there is overlap between these three
categories, but there must be some organizational way to more
easily refer to these variables without having to research for their
definitions when reading later sections of the paper.

---

> ### Author Response · Authors · 2020-11-25
> **Response to Reviewer 1**
>
> We thank the reviewer for the detailed comments and insightful questions.
> We provide our responses below:
>
> **[Exponential number of graphs for Checkmate]**
> >"In Checkmate, changes in operator implementation induce a different computation graph" <- While this is technically true, only the cost associated with that operator changes, yes? In this way, Checkmate could run multiple passes over the computed static graph with different operator costs, but this approach would require an expoential (in the number of operators with varying costs) number of evaluations.]
>
> Yes, in Checkmate, changes in operator implementation will change the cost associated with the operator. It may also change computational dependencies, and thus decisions about whether another operator's memory should be freed or not.
> Checkmate would require an exponential number of graphs to address these requirements.
>
> Thanks for bringing up this point. We also added a discussion on operator selection for Checkmate in Appendix F.
>
> ### Implementation Details
> **[PyTorch version]**
> We use PyTorch v1.5.1 for both MONeT and Checkmate.
> ---
> **[Checkmate implementation]**
> We use the Checkmate solver as-is to obtain Checkmate schedules.
> Since Checkmate does not provide an execution engine for PyTorch, we run the generated Checkmate schedules on our own execution framework.
> Our inference engine uses the same operator implementations for Checkmate and MONeT.
>
> We have added these details to Section 5 and Appendix D in the paper.
>
> ---
> **[Release Checkmate source with MONeT code?]**
> Yes.
> ---
> **[Solver runtime of MONeT vs. checkmate]**
> We have added Table 3, 4 in Appendix H to provide solver times to obtain close-to-optimal solutions for MONeT and Checkmate on multiple models and memory budgets.
> We observed that for larger models, MONeT's solver converged to a 5% close-to-optimal solution up to 27x faster, and to a 2% close-to-optimal solution up to 16x faster than Checkmate.
>
> Note that running a solver is a one-time cost for a model - once a MONeT schedule has been solved, it can be used by everyone to train the model for different purposes with different batch sizes.
> The cost (typically seconds to hours) is tiny compared to the efforts and costs to develop a model for distribution in most cases.
> ---
> **[Runtime of open-source solvers vs. commercial Gurobi solver]**
> We observe that the Gurobi solver was much faster compared to open-source solvers (eg. Cbc).
> Compared to the Cbc solver, Gurobi was 244x-259x faster in solving VGG-16 and MobileNet-V2 schedules with memory ratio 0.67.
> We think this might be due to better heuristics and pre-solving in Gurobi.
> Gurobi's license is available for free for academic use, making it more accessible than any other commercial solver behind a paywall.
> Nonetheless, we implement the MONeT solver using the CVXPY modeling language, which can seamlessly switch between Gurobi, Cbc, and other solvers if needed.
>
> ### Answer to Clarification Questions
> **[Paragraph "We measure the empirical performance ... difference of 1-2%."; is this to say 24 hours were included in execution time?]**
> We have reworded the paragraph (5th paragraph of Section 5) in the paper to make it clearer.
> The up to 24-hour solver time is not included in the execution time.
> ---
> **[Does PyTorch exceed device memory for VGG-16? Does this cause negative overhead?]**
> PyTorch does not exceed the device-memory for the VGG-16 execution in the ablation study.
> MONeT achieves a slightly faster execution time because it stores fewer tensors, saving space for choosing faster but more workspace-memory-hungry convolution algorithms.
> This effect is particularly apparent for VGG-16, which uses very large tensors in the earlier layers and its convolutional layers have extreme memory-speed tradeoffs.
> For example, a convolutions algorithm may require a large workspace memory of 2.3GB-4.5GB, while being significantly faster (by 43ms-53ms, or 3-3.8% of PyTorch training time) compared to other algorithms.
>
> The runtime gains due to convolution selection dominate over the runtime loss due to recomputations and intermediate-activated operator optimization, finally resulting in a net runtime improvement.
>
> ### Comments Regarding Notation
> We appreciate the reviewer's insightful feedback.
> We have incorporated the suggestions in the paper.
> The change includes an extended notation table in Appendix A that additionally describes variable y, the sets L, D, and S among other variables.
> The variable r (the reviewer's understanding is correct) is also mentioned in the table.
> We also refer readers to this table early in our paper for ease of reading.
>
> **[Standardized notation by the three categories]**
> Great idea! We added explanations regarding our notation convention along with our notation table in Appendix A.
> In particular, notations with only i in subscript/superscript relate to the forward pass, with only k relate to the backward pass, and with both i and k relate to the recomputation phase.

---

### Official Review · AnonReviewer4 · 2020-10-28
**Valuable contribution for deep learning training**

**Rating:** 8
**Confidence:** 4

**Review:**

Training deep learning models is becoming increasingly challenging due to a memory bottleneck that limits the size of the feature maps that can be stored. The paper presents an automatic framework (MONET) that minimizes the memory footprint for deep networks. The novelty of MONET is that it jointly optimizes over: (a) global compute graph level techniques (such as checkpointing) and (b) local techniques (such as memory-efficient implementations of individual operators). While there are several existing works that focus separately on optimizing global techniques (e.g. the work on “Checkmate”) or local techniques, MONET is the first to jointly optimize over global and local techniques.

The memory constraints are carefully analyzed for the forward and backward passes, and expressed as a 0-1 integer program, which is then solved using the state-of-the-art solver Gurobi. The experimental evaluation confirms the theoretical gains provided by the solution of the optimization problem. In particular, the authors compare with a vanilla implementation in PyTorch, with the Checkmate-D (default Checkmate that uses global techniques), and with Checkmate-O (post-optimized to greedily run the fastest convolution algorithm). It is interesting to notice that MONET offers significant gains in all cases, even over Checkmate-O, underlying the need to perform the joint optimization in order to obtain the best schedule.

I advocate for the acceptance of the paper. The memory bottleneck is an acute problem in deep learning, and the paper provides a practical solution that can alleviate this problem to some extent. I encourage this line of work and hope to see such schedule optimization become a standard option in deep learning frameworks. The paper is very well written, with a clear description of the approach and convincing experimental evaluation, while providing abundant references about related work.

Minor comments, questions for the authors:
The memory requirements for some of the networks shown in the experimental evaluation are still small. While I understand that the GPU memory is limited to 16GB and there was a desire to compare against vanilla implementations, I think it would be interesting to show how models that require a lot of memory for training can benefit from MONET. For example, the UNet that was used in the paper seems to be the 2D version. The 3D UNet can require up to several hundred GB without optimizations (for varying batch sizes). Would it be possible to include even a simple estimate of the possible gains when using MONET? What would be the computational tradeoff to train it on the GPU used in the paper?

---

> ### Author Response · Authors · 2020-11-25
> **Response to Reviewer 4**
>
> We thank the reviewer for the review and appreciate their encouragement towards this line of work.
>
> We appreciate the great suggestion regarding 3D-UNet; we have added the results to Appendix J.
> As a quick summary of the results: using MONeT, we could run a 3D-UNet model using 0.54x times the memory and with 8.86% compute overhead over PyTorch. We provide a detailed runtime-memory tradeoff curve for MONeT below (from Appendix J). At a 0.81 memory ratio, we incur almost no computational overhead, because MONeT, making use of operator optimizations, is able to bring down the recomputation cost to zero.
>
> (We used the 3D-UNet model present at https://github.com/wolny/pytorch-3dunet with a batch size of 22, which just fits on a 16 GB P100 GPU. Other training configuration is similar to "3DUnet\_confocal\_boundary" provided in the repository.)
>
>
> | Memory ratio | Runtime overhead |
> |-------------:|-----------------:|
> |         0.54 |            8.86% |
> |         0.58 |            5.14% |
> |         0.65 |            1.81% |
> |         0.72 |            0.78% |
> |         0.81 |            0.20% |
> |              |                  |

---

### Official Review · AnonReviewer2 · 2020-10-29
**The paper proposes MONeT to jointly optimizing both local operators and the global compute graph. The authors express this joint optimization problem as a integer program. Results are impressive with 3x memory reduction at a small overhead or 1.2-1.8x reduction at the same computation.**

**Rating:** 6
**Confidence:** 3

**Review:**

Optimizing Neural Network Memory is broadly done through two channels - 1)  local operator level - like storing only signs of a ReLU function or bit quantization
2) global graph level - like optimizing checkpointing schedule for a given compute graph. These channels are usually orthogonal/independent.

In this paper, the authors propose MONet which tries to find the best checkpointing schedule that can jointly optimize both the above channels.
The authors create an auxiliary graph to encapsulate operators and perform schedule optimization on the new graph rather than the usual graph in existing frameworks.

To obtain the new graph, they first perform a theoretical quantification of the peak memory consumption of the forward pass,backward pass and recomputation.
Then, under a fixed memory budget M, we try to optimize the operators for computational efficiency.

Each operator has multiple implementations that trade-off workspace memory 'c' and compute efficiency \tau and we can exactly use only one implementation per operator.
So we end up with a linear combination of compute costs for all operations that looks like

\sum_i\sum_l(\tau_i^l \delta_il) where \delta is an indicator of which implementation to choose.

With 3 such double summations, the authors come up with a join objective function which constraints the overall memory sum to <=M
and minimizes the compute cost objective above (which can be solved by a linear program).

On several popular architectures like ResNet-50, VGG-16 and UNet etc, MONeT outperforms Checkmate by abt 1.3-2x on memory and full-blown PyTorch by abt 2-3x.

Pros:
1. Thoretically optimized
2. Experiments on multiple CNN architectures

Cons:

1. Effetcively no comparisons against any other schemes except Checkmate.
2. No other architectures optimizations like Transformers or large scale FFNs.

---

> ### Author Response · Authors · 2020-11-25
> **Response to Reviewer 2**
>
> We thank the reviewer for the comments and feedback.
> We provide our responses below:
>
> **[Comparison with schemes apart from Checkmate]**
> We have added Appendix K which provides a comparison between MONeT and an operator-based memory-efficient scheme Gist [1].
> Gist uses various techniques to encode stashed forward pass tensors into smaller tensors which require less memory.
> We find that MONeT uses 1.4x-2.1x less memory than Gist for multiple architectures, while incurring lower compute overheads.
>
> We evaluate Gist on PyTorch using MONeT’s execution framework, since an official implementation of Gist is not available.
> We get similar memory saving results for reimplemented-Gist as in the original paper for VGG-16, but see a higher computational overhead in our implementation of Gist.
> This could be because of evaluations on different frameworks (PyTorch v/s CNTK) and different GPU models (Nvidia P100 v/s Nvidia Maxwell GTX Titan X).
> To ensure a fair comparison, we just compare the maximum memory savings obtained by MONeT against our Gist implementation, and report the compute overhead for completeness.
>
> Appendix K provides further details of the implementation and comparison with Gist.
>
> We are also very happy to take suggestions for other baselines to compare against.
>
> [1] Animesh Jain, Amar Phanishayee, Jason Mars, Lingjia Tang, and Gennady Pekhimenko. Gist: Efficient data encoding for deep neural network training. In ISCA, 2018.
>
> ---
>
> **[Optimizations on other architectures]**
> Great comment!
> We have added additional results on a memory-intensive model, 3D-UNet, in Appendix J.
> We observe a consistent memory reduction to 0.54x of PyTorch memory with an overhead of 8.86% in the high-memory regime.
> Together with other results we present, we've observed consistent and good memory-overhead trade-offs on 6 models, with a wide range of memory budgets.
> We are working on experimenting with even more models, including Transformers.

---

### Author Response · Authors · 2020-11-25
**Thanks for the reviews and summary of main changes**

We thank all reviewers for their time and valuable feedback.
We are glad that all reviewers recommend acceptance and find the results impressive and convincing (R1, R2, R4), the description clear and well-written (R1, R4), and like the integer-programming formulation (R1, R3).

According to the comments, we have updated the paper with the following changes:

- Clarified a paragraph talking about solver times in the paper (5th paragraph of Section 5)

- Added more notations regarding the formulation in Appendix A

- Added implementation details about Checkmate in Appendix D

- Added Appendix H which provides a discussion about solver times and ILP statistics

- Added Appendix J which provides the runtime-memory tradeoff results for 3D-UNet using MONeT

- Added Appendix K which provides a comparison of MONeT with a data encoding scheme, Gist

We answer each reviewer's questions in an individual response for each review below.

---

### Decision · Program_Chairs · 2021-01-07
**Final Decision**

**Decision:**

Accept (Spotlight)

**Comment:**

The reviewers all agree that Monet proposed in the paper which optimizes for both local and global memory saving in Deep learning models is theoretically sound and experimentally convincing.
Accept!